# Efficacy of targeted indoor residual spraying with the pyrrole insecticide chlorfenapyr against pyrethroid-resistant *Aedes aegypti*

**Azael Che-Mendoza**[1], **Gabriela González-Olvera**[1], **Anuar Medina-Barreiro**[1], **Carlos Arisqueta-Chablé**[1], **Wilberth Bibiano-Marin**[1], **Fabián Correa-Morales**[2], **Oscar D. Kirstein**[3], **Pablo Manrique-Saide**[1], **Gonzalo M. Vazquez-Prokopec**[3]*

**1** Unidad Colaborativa para Bioensayos Entomologicos, Universidad Autonoma de Yucatan, Merida, Yucatan, Mexico, **2** Centro Nacional de Programas Preventivos y Control de Enfermedades (CENAPRECE) Secretaria de Salud Mexico, Ciudad de Mexico, Mexico, **3** Department of Environmental Sciences, Emory University, Atlanta, Georgia, United States of America

* gmvazqu@emory.edu

## Abstract

### Background

There is an increased need to mitigate the emergence of insecticide resistance and incorporate new formulations and modes of application to control the urban vector *Aedes aegypti*. Most research and development of insecticide formulations for the control of *Ae. aegypti* has focused on their peridomestic use as truck-mounted ULV-sprays or thermal fogs despite the widespread knowledge that most resting *Ae. aegypti* are found indoors. A recent modification of indoor residual spraying (IRS), termed targeted IRS (TIRS) works by restricting applications to 1.5 m down to the floor and on key *Ae. aegypti* resting sites (under furniture). TIRS also opens the possibility of evaluating novel residual insecticide formulations currently being developed for malaria IRS.

### Methods

We evaluated the residual efficacy of chlorfenapyr, formulated as Sylando 240SC, for 12 months on free-flying field-derived pyrethroid-resistant *Ae. aegypti* using a novel experimental house design in Merida, Mexico. On a monthly basis, 600 female *Ae. aegypti* were released into the houses and left indoors with access to sugar solution for 24 hours. After the exposure period, dead and alive mosquitoes were counted in houses treated with chlorfenapyr as well as untreated control houses to calculate 24-h mortality. An evaluation for these exposed cohorts of surviving mosquitoes was extended up to seven days under laboratory conditions to quantify "delayed mortality".

### Results

Mean acute (24-h) mortality of pyrethroid-resistant *Ae. aegypti* ranged 80–97% over 5 months, dropping below 30% after 7 months post-TIRS. If delayed mortality was considered (quantifying mosquito mortality up to 7 days after exposure), residual efficacy was above

**Data Availability Statement:** Data are available at Mendeley Data: Vazquez-Prokopec, Gonzalo

(2021), "Efficacy of the pyrrole insecticide chlorfenapyr against pyrethroid resistant Aedes aegypti", Mendeley Data, V1, doi: 10.17632/8f6w3vj74t.1.

**Funding:** This project received support from Innovative Vector Control Consortium (Award ID:48835), Emory Global Health Institute and Marcus Foundation (00052002), and partly by the National Institutes of Health, National Institute of Allergy and Infectious Disease (U01AI148069; Vazquez-Prokopec, PI). The funders had no role in study design, data collection and analysis, decision to publish, or preparation of the manuscript.

**Competing interests:** The authors have declared that no competing interests exist.

90% for up to 7 months post-TIRS application. Generalized Additive Mixed Models quantified a residual efficacy of chlorfenapyr of 225 days (ca. 7.5 months).

## Conclusions

Chlorfenapyr represents a new option for TIRS control of *Ae. aegypti* in urban areas, providing a highly-effective time of protection against indoor *Ae. aegypti* females of up to 7 months.

---

## Author summary

Vector control (VC) for managing *Aedes aegypti* and reducing transmission of *Aedes*-borne diseases is largely focused on peridomestic insecticide applications. However, the indoor resting behavior of *Ae. aegypti* and the acceleration of insecticide resistance owed to reduced modes of action have diminished the effectiveness of many VC tools. A targeted Indoor residual spraying (TIRS) modality in experimental housing units was employed to investigate the potential of chlorfenapyr, a pyrrole-class insecticide with known effectiveness to resistant mosquito species. This was the first investigation for chlorfenapyr use against locally resistant *Ae. aegypti* (Merida, Mexico) with this approach. Two treatment arms were investigated in the present study: TIRS and a control house where only water was sprayed. A comparison of entomological efficacy for TIRS applied to interior perimeter walls below 1.5 m with chlorfenapyr (formulated as Sylando 240SC) at 250 mg/m$^2$ over 12 months was assessed. TIRS chlorfenapyr treatments were highly efficacious and led to acute mortalities (after 24 exposure) above 80% up to 5 months; delayed mortalities (to *Ae. aegypti*) were monitored over seven days post exposures vs untreated controls. When delayed mortality was considered, residual efficacy of chlorfenapyr extended to 7 months. These data provide evidence that TIRS chlorfenapyr is an effective *Aedes* management tool that surpassed efficacy profiles for other TIRS insecticides that have been previously reported with this method. Further, Chlorfenapyr emerges as a novel addition to *Ae. aegypti* VC, and future studies should focus on its effectiveness and residual power as part of Phase II-III TIRS trials.

## Introduction

Controlling the anthrophilic disease vector *Aedes aegypti* has long been conducted by peridomestic application of truck-mounted ultra-low volume spraying, thermal fogging and larviciding [1,2]. Adult female *Ae. aegypti* are typically found indoors in urban settings, where they feed frequently and almost exclusively on human blood [3–5] and rest on surfaces that are unreachable with the routinely used insecticide methods. Peridomestic mosquito control tactics, therefore, lead to poorly-efficient and in the best case, transient control of the epidemiologically important biting female mosquitoes (e.g., [5]) and thus, with limited impact in preventing arboviral disease transmission [6].

A novel application technique, which exploits *Aedes aegypti* resting behavior, termed targeted indoor residual spraying (TIRS), focuses the selective application of residual insecticides in lower walls (<1.5m) and other primary *Ae. aegypti* resting locations (under beds and furniture), reducing insecticide volumes and treatment time [7,8]. The development of TIRS was rooted on prior success in controlling *Ae. aegypti* using perifocal spraying of DDT [6,9,10],

and recent evaluations in a novel experimental house setting in Merida, Mexico [7]. Effectiveness of TIRS implementation has been confirmed in Cairns, Australia, where coverages of 60% or more led to reductions in dengue virus incidence of >86% [8]. Furthermore, modeling studies indicate that the highest effectiveness of TIRS occurs when the method is deployed preventively prior to the regular transmission season, instead of reactively to cases [11–13]. Preventive TIRS, while considered an approach that can overcome the limitations of IRS and increase insecticide application effectiveness, is dependent on having insecticide molecules to which *Ae. aegypti* is susceptible and insecticide formulations that can provide sustained control for 5 months or more [12].

Recent advancements in new and repurposed chemistry to mitigate mosquito-borne diseases have been seen from the development of non-pyrethroid IRS formulations to control malaria vectors [14–18]. Some of the innovation in new molecules stands from their unique toxicity mechanisms, which rely more on mosquito physiology than on "usual" neurological or simple detoxification pathways. Chlorfenapyr (commercially available as Phantom Termiticide -Insecticide in the United States, BASF for urban pest control and Sylando 240SC, BASF for public health use) is a new insecticide class (pyrrole) that acts as a physiological toxin, requiring activation as a pro-insecticide [19,20] to exert mosquito mortality [19]. Chlorfenapyr is a halogenated pyrrole that uncouples oxidative phosphorylation processes in mitochondria [20]; in other words, affects insect's ability to produce energy in their mitochondria which consequently affects crucial and vital functions until eventual death. The mode of action of chlorfenapyr on an insect's metabolism is particularly relevant for the control of vectors harboring metabolic insecticide resistance mechanisms (e.g., cytochrome P450, glutathione S-transferases), as increased metabolic activity increases the activation of the toxin and increase mosquito mortality [19]. Furthermore, as these new physiological insecticides depend on the mosquito metabolism to act, they generally present delayed toxicity (in the order of 1–5 days) when insects are inactive or constrained to a cage, making their evaluation using conventional neuro-toxic tests (e.g., WHO cone bioassay) challenging [21]. While chlorfenapyr has recently been evaluated against Anopheles sp. Vectors, no rigorous evaluation of its efficacy on *Ae. aegypti* has been published.

TIRS evaluation of the carbamate Bendiocarb on a novel experimental house setting established in Merida, Mexico (i.e., typical residential houses rented long-term and double-screened to allow for free-flying mosquitoes to be exposed to diverse insecticide treatments), led to a 4-month residual efficacy against pyrethroid-resistant *Ae. aegypti* [22]. Such experimental setup provides a unique opportunity to evaluate new insecticide formulations for TIRS against *Ae. aegypti*, as it saves the cost of running experiments in the open field or in expensive lab enclosures. Here, we evaluated the residual efficacy of chlorfenapyr (Sylando 240SC; BASF) against a locally-derived, pyrethroid-resistant, strain of *Ae. aegypti*.

## Methods

### Ethics statement

This was an experimental study, and because mosquitoes were released into uninhabited houses rented on long-term contracts, we did not require an Institutional Review Board evaluation.

### Experimental house layout

We conducted this evaluation within two experimental houses located in Ciudad Caucel, a neighborhood of the subtropical city of Mérida, México [22]. The houses are rented long-term by the Universidad Autónoma de Yucatán (UADY) after explaining the purpose and extent of

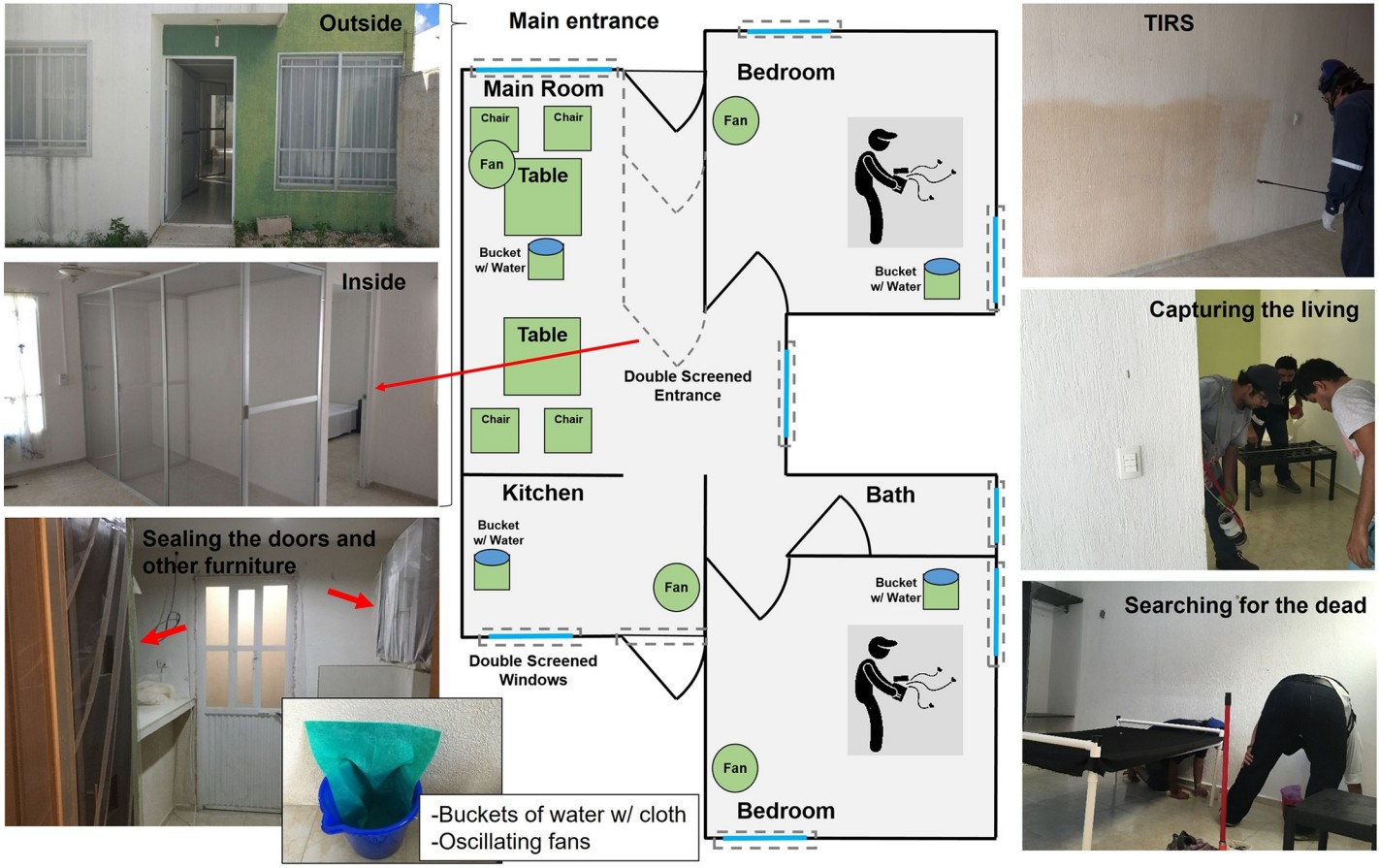

**Fig 1. Experimental houses from Merida, Mexico.** Two houses with similar area (size: 144.5 ± 7.12 m3) and layout were used. Windows and doors (inside & outside), including furniture were sealed. A double screened entrance was also installed. Simulated furniture was standardized in each house. Buckets of water with cloth piece and oscillating fans were installed to keep optimal humidity and temperature as showed in the figure.

the study to their owners. Distance between experimental houses was 0.3 km. The houses were similar in floor plan and design; all were concrete, single-story and had a living-dining room, two bedrooms, one bathroom and one kitchen (Fig 1). Houses were on average 57.8 ± 2.8 m$^2$ (mean ± SEM) and uniformly had walls 2.5 m in height. Construction characteristics were that of subsidized middle to low-income housing in Mérida, typical of areas with high ABD transmission [23].

To prevent any mosquitoes used in the experiments from escaping from the houses, all windows and doors were screened on both the outside and inside of each house before the study began. Additionally, a double screened-door vestibule was built into the main entrance of each house to allow personnel to enter and exit while preventing mosquitoes from escaping (Fig 1). Sinks, drains and toilets were also sealed with window screening. Existing furniture within houses was removed, and where furniture could not be removed (*e.g.*, built-in kitchen or closet cabinets) it was sealed with window screening. Houses were then refurnished with standardized furniture and clothing that represented typical elements found within houses (Fig 1). Furniture within the living room included two black plastic tables and four plastic chairs. Within each bedroom was a bed made out of PVC tubing and black cloth, a black plastic nightstand and six articles of clothing (3 black and 3 white) hung within the closet. Four plastic buckets (1 L) were half filled with water and a dark cloth and placed throughout each house to provide

moisture into the environment and reduce mosquito mortality due to desiccation. Ant baits (Antex Gel, Allister de México with 0.05% abamectin) were placed next to each door or any other location where ants were observed to enter the experimental houses. The house layout was carefully designed to mirror elements and surface materials found in most homes but ensuring standardization in a way that allowed replication and comparability between replicates.

## Insecticide application

Insecticide and untreated controls (water) were applied within experimental houses on 18 March 2019. Manual compression sprayers, IK-Vector Control Super (Goizper Group, Antzuola, Spain) were fitted with 8002EVP nozzle and a Goizper Low Pressure Control Flow Valve (output pressure 1.5 bar) to administer sprays to houses at a flow rate of 580 mL / min ± 5%), according to following preparations: Sylando 240SC Target dose 250 mg/m$^2$ and 286 mL diluted in 7.5L water as recommended by the manufacturer in the proposed label and detailed in prior IRS trials conducted in Africa and India by WHOPES [24]. All applications were performed by the same applicator. TIRS application was conducted as described in Dunbar et al. [22]. Briefly, insecticide (or water, for the control) was applied to walls below 1.5 m and under furniture or to the undersides of furniture. Furniture was not removed from experimental houses during the insecticide application and insecticide was not applied to clothing or the plastic buckets with water.

## Mosquito strain

To test the residual efficacy of each IRS application method, groups of 100 *Ae. aegypti* females three to seven days old from F4 generation were released within each experimental house. The strain used (Juan Pablo strain, JP) was locally derived, had a high level of resistance to pyrethroids but full susceptibility to carbamates [25,26]. The JP strain was reared and maintained at the insectaries of the Unidad Colaborativa para Bioensayos Entomológicos (UCBE), UADY, Mérida, México, at constant laboratory conditions (27˚C and 60% RH). Resistance is maintained by periodic mixing of the colony with recently hatched larvae from field-collected eggs and monitored using the CDC bottle bioassay [27] and genotyped using standard PCR methods [25,26] to detect two of the most common single nucleotide polymorphisms of the voltage gated sodium channel gene (i.e., at positions 1,016 and and 1,534) as described elsewhere [25,26]. Mosquitoes released into houses had only been provided sugar solution and were non-bloodfed.

## Intervention evaluation

Post-insecticide application, mosquitoes were released into the experimental houses (both in houses treated with chlorfenapyr as well as untreated control houses) eleven times over a 12-month period; 1) +1 day, 2) +14 days, 3) +1 month, 4) +2 months, 5) +5 months, 6) +7 months, 7) +8 months, 8) +9 months, 9) +10 months, 10) + 11 months, and 11) +12 months (see Table 1). Replication of this design occurred by conducting three independent releases, on three consecutive days, for each period (Fig 2). To facilitate mosquito detection, all experimental houses were vacuumed and swept clean of any debris on the floor one day prior to mosquito release. After a 24 hr exposure in the houses, a team of four field technicians entered each house and searched for live mosquitoes using a Prokopack aspirator [28] and searched by hand for dead mosquitoes. This 24-h exposure period allowed quantification of acute mortality. Searching for *Ae. aegypti* ceased when either 100 mosquitoes were collected or > 20 minutes elapsed after the last mosquito was collected (circa 30–40 min / house). Sampling dates are provided in Fig 2 for release of cohorts into experimental houses. Acute (24-h) mortality

**Table 1. Sampling dates for Release of Mosquitoes into experimental houses in Ciudad Caucel neighborhood of Merida, MX.** For each of the two experimental houses, three consecutive releasing events were implemented in each period of time to evaluate, using days as replicates (also see Fig 2). The mosquito strain (Juan Pablo) was pyrethroid-resistant; resistance was maintained by periodic reseeding of populations with field-collected eggs (see methods).

| Post-application releasing | Days post application | Releasing dates | Number of Mosquitoes |
|---|---|---|---|
| 1 day | 1 | 19–21 March, 2019 | n = 600 |
| 2 weeks | 14 | 2–4 April, 2019 | n = 600 |
| 1 month | 30 | 16–18 April, 2019 | n = 600 |
| 2 months | 60 | 27–29 May, 2019 | n = 600 |
| 5 months | 150 | 11–13 August, 2019 | n = 600 |
| 7 months | 210 | 21–23 October, 2019 | n = 600 |
| 8 months | 240 | 20–22 November, 2019 | n = 600 |
| 9 months | 270 | 15–17 December, 2019 | n = 600 |
| 10 months | 300 | 19–21 January, 2020 | n = 600 |
| 11 months | 330 | 16–18 February, 2020 | n = 600 |
| 12 months | 360 | 17–19 March, 2020 | n = 600 |

was calculated from the number of dead/live *Ae. aegypti* found at the end of the exposure period in the houses. Exposed mosquitoes were held at the UCBE insectary inside bugdorm cages (30x30x30 cm) for 7 days at 26 ± 2°C and 75 ± 5%RH and monitored daily for signs of intoxication to quantify "delayed mortality" because uncoupling of oxidative phosphorylation and the necessary requirement for mosquitoes to enzymatically convert parent chlorfenapyr (CL303630) to its n-dealkylated metabolite (CL303268) delay the appearance of toxicity effects in mosquitoes [19,29]. On each house, we placed three unsprayed control cups (250 mL) containing 10 JP strain females each during the 24-h exposure period to have an independent measure of mosquito mortality due to the temperature and humidity conditions of the experimental houses. This measure was estimated at the +5, +7, +8–12 months post-application evaluations, which coincided with the warmest periods of the year in Merida.

## Statistical analyses

For each sampling period, acute and delayed mortalities were calculated per house by dividing the number of dead individuals by the number of individuals released. Missing individuals were assumed to be dead. Due to the mortalities in the control group (when they were observed) which ranged from 2–23%, the mortality calculation was corrected according to the

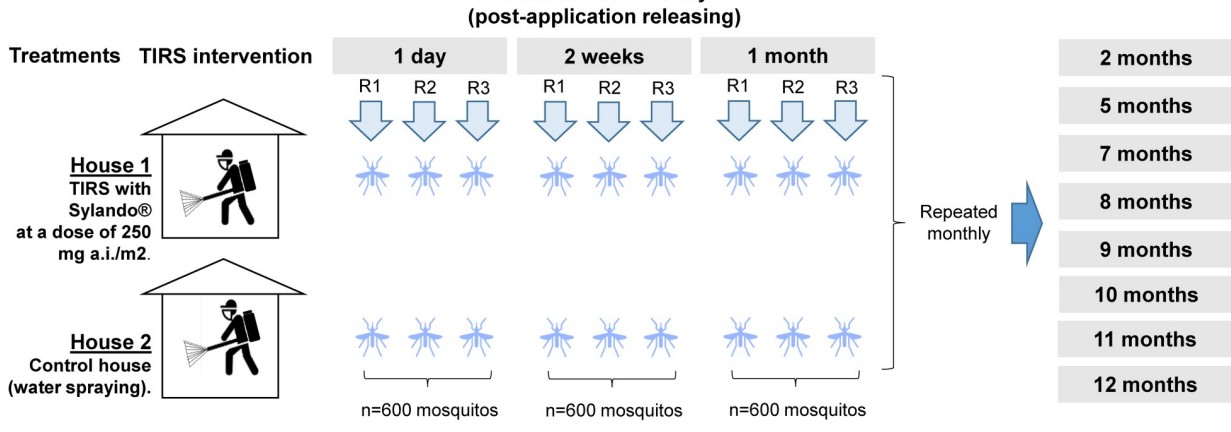

**Fig 2. Study design.** Each arrow represents a releasing event. R = replicate.

formula of Abbott (1925). On each evaluation date, corrected acute mortalities were compared to the 80% threshold set by WHO as the cutoff for effective insecticidal effect of indoor residual spraying [30]. Further, both acute and delayed mortalities were compared between treatment and control using binomial generalized linear mixed models (GLMM) in R 4.0.5 statistical software (https://www.r-project.org/) using package lme4. For each date, treatment was classified as fixed effect and experimental replicate was classified as a random effect.

A Generalized Additive Mixed Model (GAMM) determined the association between acute and delayed mortality and the time (in days) since TIRS application. Time to intervention was calculated by estimating the number of days that elapsed between TIRS and the entomological evaluation. The full model had the form: Mortality = $\alpha$+ f(Days) + Z(Replicate)+$\varepsilon$. Where Z (Replicate), represents a random effects term associated with observations from the same time point, $\alpha$ the model constant and $\varepsilon$ the error term. We fitted f(Days), the non-linear term of mortality and days since TIRS, by applying a penalized cubic spline function to the data and a Gaussian link function to fit the model. The parameter f(Days) was fitted separately to the control and chlorfenapyr data. Exploration of fitted f(Days) allowed assessing the temporal trend in *Ae. aegypti* mortality after TIRS. Specifically, since f(Days) describes the non-linear fit of the time since TIRS application to the mortality data, we used the parameter's 95% credible interval (95%CI) to quantify: 1) if the 95%CI of f(Days) differed significantly between control and chlorfenapyr treatments; and 2) at what time point the predicted non-linear fit for chlorfenapyr (with 95% CI) went from positive to negative, indicating a loss of impact of the insecticide on mosquito mortality. The package mgcv was used to fit and plot the results of the GAMM.

## Results

TIRS was implemented according to standard protocol (spraying walls below 1.5 m and under furniture) on March 18, 2019. A total of 7,200 *Ae. aegypti* females were released within the experimental houses throughout the trial. Recapture of released mosquitoes (dead and alive) averaged 97.5 ± 5.3% (Mean ± SEM; n = 66 releases). Based on prior studies applying TIRS, we attribute high recovery to pre-cleaning the floors of experimental houses the day before mosquitoes were released and to effective management of ants using baits. Mortality within cups left inside houses to monitor natural mortality averaged 3.2 ± 1.1, 4.8 ± 0.8, 2.3 ± 1.5, 1.9 ± 0.6, 4.4 ± 1.3%, 1.5 ± 0.7% and 5.0 ± 1.7% (Mean ± SEM) for evaluations from +5, +7, +8, +9, +10, +11 and +12 months post-application, respectively, indicating negligible effect of high summer temperatures on mortality. Before the first release, recently emerged female *Ae. aegypti* mosquitos were tested for susceptibility to permethrin, deltamethrin and chlorpyrifos (100 females per insecticide). At the diagnostic time for each insecticide, 72%, 94% and 100% female mosquitoes were dead in the permethrin, deltamethrin, and chlorpyrifos groups, respectively. After 6 months, and to maintain genetic diversity and resistance mechanisms, the laboratory strain was mixed with a batch of 1,000 recently emerged larvae from field-collected eggs. Emerging adults from the mixed colony experienced mortalities at the diagnostic time of 62% for permethrin, 92% for deltamethrin and 100% for chlorpyrifos. A subsample of 141 female *Ae. aegypti* from the mixed colony was genotyped for the presence of the two most common kdr mutations. For the 1,016 mutation, 27.7% mosquitoes were homozygous susceptible, whereas 26.2% were homozygous resistant and 46% were heterozygous. For the 1,534 mutation, only 10.6% were homozygous susceptible, whereas 66.0% were homozygous resistant and 23.4% heterozygous. This information is indicative of pyrethroid resistance in the population.

Acute mortality of female *Ae. aegypti* released into the houses was significantly higher and sustained in houses sprayed with chlorfenapyr compared to control houses up to 11 months

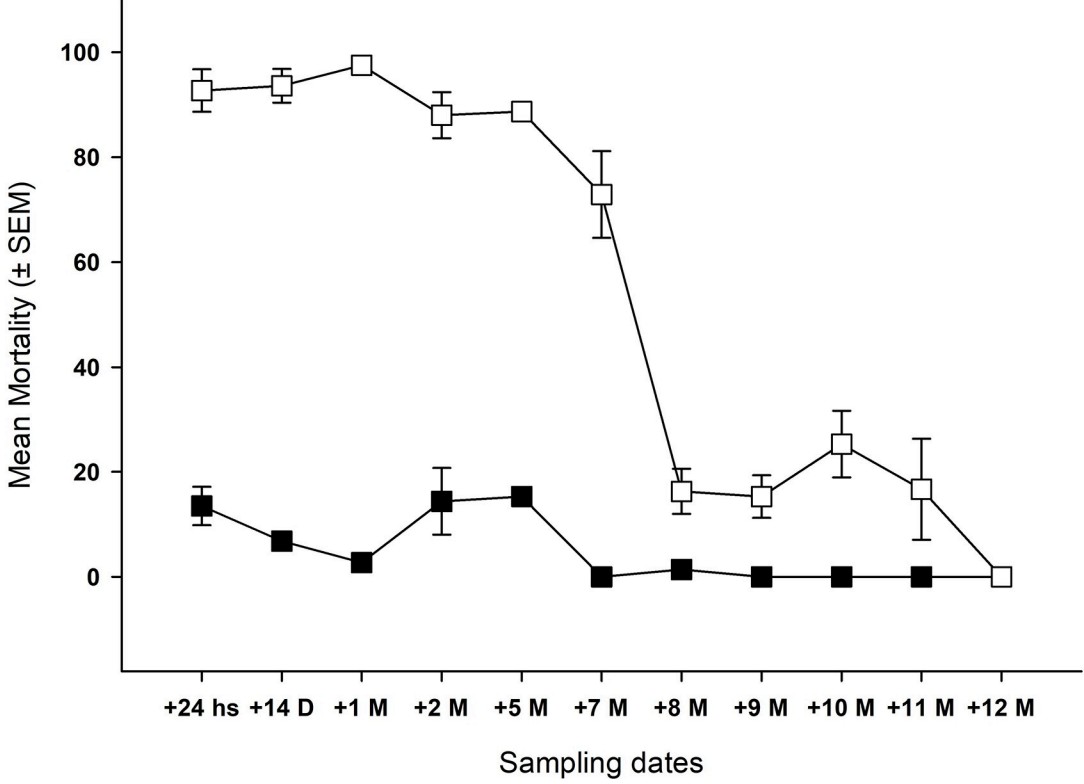

**Fig 3. Mortality of pyrethroid-resistant *Ae. aegypti* (Juan Pablo Strain) by TIRS application method using chlorfenapyr (Sylando 240SC) over time.** Mean (± the standard error of the mean, SEM) of acute (24-h) corrected mortality (Abbott, 1925) of pyrethroid resistant *Ae. aegypti* (Juan Pablo strain) by TIRS using Sylando 240SC formulation 250 mg a.i./m² (white squares) compared to a control treated with water (black squares).

post spraying (Fig 3, Table 2). Abbott-corrected average mortalities (including their standard error) were equal on higher than the 80% mortality threshold up to 5 months post-TIRS. A remarkable reduction on the mortality (15–16%) was observed at 8 to 11 months, whereas no mortality was observed at 12 months (Fig 3, Table 2).

Delayed mortality was recorded during 7 days post-exposure for most collection periods (Fig 4). Total delayed mortality (100%) was observed after 48 hr of exposure at 1 and 2 months post-TIRS application. At 5 and 7 months post TIRS application, delayed mortality was 96.6% and 99.3% after 2–7 days of observation, respectively. At 8 and 9 months the delayed mortality reached 75% and 64% after 7 days of observation respectively. At 10 & 11 months the maximum mortality reached after 7 days of observation was 41% and 36% respectively. At 12 months no delayed mortality was observed (0% after 7 days of observation). Similar levels of statistical significance as described for acute mortality when comparing chlorfenapyr and control data were observed for delayed mortality (S1 Table).

Fig 5 shows the plot of f(Days), obtained after fitting a GAMM to the mortality data of the control and chlorfenapyr houses. The y-axis can be interpreted as the effect of time since TIRS on mosquito mortality. When the predicted value and its 95% credible interval are negative, it means that there is a significant reduction in mortality. Chlorfenapyr led to a significant reduction in mortality up to 225 days (ca. 7.5 months, vertical line on right panel of Fig 5A and 5B) post-TIRS application.

**Table 2. Average (min-max) raw acute (24-h) mortality data and Abbott- corrected mortality throughout the 11 sample periods (24 hours, 2 weeks, 1, 2, 5, 7, 8, 9, 10, 11 and 12 months) and results from a Generalized Linear Mixed Model (GLMM) quantifying the significance in mortality between control and treatment measures (control used as baseline).**

| Days post TIRS | Treatments | Percent (range between replicates) | | | GLMM | |
| | | Recapture after 24-h | Mortality | Corrected Mortality | Coefficient (std. error) | P-value |
|---|---|---|---|---|---|---|
| 1 | Chlorfenapyr | 87.3 (82–94) | 93.4 (87–100) | 92.7 (86–100) | 0.792 (0.05) | **0.0001** |
| | Control | 92 (88–94) | 13.5 (6.7–19.1) | — | | |
| 14 | Chlorfenapyr | 92 (86–96) | 93.9 (87.8–98) | 93.6 (87.2–97.8) | 0.868 (0.04) | **<0.0001** |
| | Control | 96.7 (94–98) | 6.8 (4–10.2) | — | | |
| 30 | Chlorfenapyr | 100 | 97.6 (95.2–100) | 97.5 (95.2–100) | 0.949 (0.02) | **<0.0001** |
| | Control | 100 | 2.7 (0–4) | — | | |
| 60 | Chlorfenapyr | 96.7 (94–100) | 93.1 (87.2–97.3) | 82 (77.3–88.7) | 0.677 (0.07) | **0.0007** |
| | Control | 92.7 (78–100) | 14.4 (2–23.1) | — | | |
| 150 | Chlorfenapyr | 93.3 (90–96) | 90.4 (88.2–93.1) | 88.6 (86–91.8) | 0.734 (0.02) | **<0.0001** |
| | Control | 100 | 15.3 (14–16) | — | | |
| 210 | Chlorfenapyr | 100 | 72.9 (56.5–83.3) | 72.9 (56.5–83.3) | 0.730 (0.008) | **0.0009** |
| | Control | 100 | 0 | — | | |
| 240 | Chlorfenapyr | 100 | 17.3 (12–28) | 16.3 (12–24.9) | 0.149 (0.05) | **0.0298** |
| | Control | 98.7 (96–100) | 1.4 (0–4.2) | — | | |
| 270 | Chlorfenapyr | 100 | 15.3 (8–22) | 15.3 (8–22) | 0.153 (0.05) | **0.0194** |
| | Control | 100 | 0 (0–0) | — | | |
| 300 | Chlorfenapyr | 100 | 25.3 (18–38) | 25.3 (18–38) | 0.253 (0.06) | **0.0164** |
| | Control | 100 | 0 (0–0) | — | | |
| 330 | Chlorfenapyr | 100 | 16.7 (14–18) | 16.7 (14–18) | 0.166 (0.01) | **0.0002** |
| | Control | 100 | 0 (0–0) | — | | |
| 360 | Chlorfenapyr | 100 | 0 | 0 | N/A | 1 |
| | Control | 100 | 0 (0–0) | — | | |

## Discussion

This study provides information about a new insecticide chemistry for the urban control of pyrethroid-resistant *Ae. aegypti* using a novel experimental house system that incorporates typical living conditions in urban areas of an endemic area for ABVs. Results from this study show that a single TIRS application of chlorfenapyr (Sylando, 240SC Target dose 250 mg/m²) led to mosquito mortalities above 80% for up to 5 months and to delayed mortalities above the 80% threshold for up to 7 months. Operationally, results suggest that a single application of chlorfenapyr can provide a new highly-effective and sustainable alternative for TIRS application for ministry of health institutional programs to control *Ae. aegypti* in urban areas.

Studies both in the laboratory and field environments have shown the ability of many insect species to rest on surfaces treated with chlorfenapyr for extended periods of time [31–33]. The non-repellent nature of chlorfenapyr, described in other studies on mosquitoes [21,34–36], may have led to greater resting times and insecticide uptake compared to pyrethroids, contributing to observed mortalities in experimental houses. The physiological effect of chlorfenapyr on free-flying mosquitoes may have also contributed to the extended and significant direct and delayed mortality effects observed. The enzymatic transformation of parent chlorfenapyr (CL303630) to its pro-insecticidal metabolite (CL303268) can be slow and quite variable, but generally unidirectional once conversion has started [29]. The uncoupling of oxidative phosphorylation can be influenced by many exogenous and endogenous factors: temperature, cuticular penetrations, physical movement of challenged insects, host-seeking behaviors,

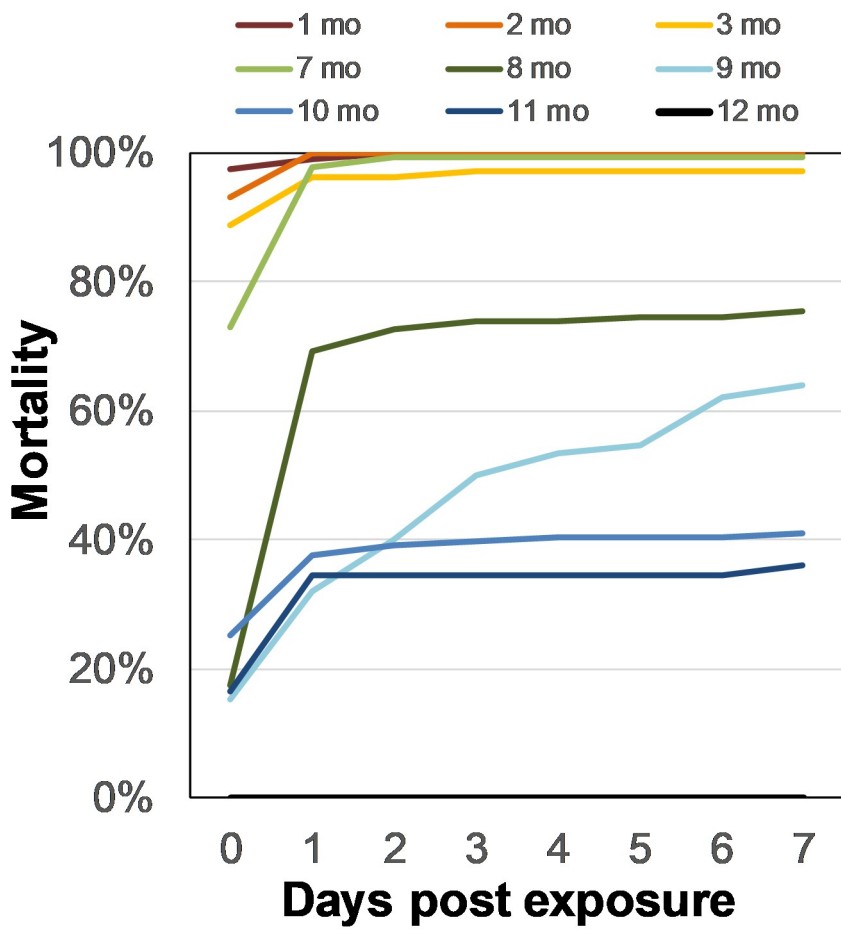

**Fig 4. Cumulative delayed mortality at 1 to 12 months post-application (up to seven days after exposure to Sylando 240SC applied at 250 mg/m² via TIRS).** The mortality at 24 h of collecting is represented as "0". For 12 mo, mortality was 0 throughout the evaluation.

blood-feeding status of mosquitoes, concentrations of chlorfenapyr challenged to insects from different substrates, degree of metabolic activity already within target pests and antagonisms by known metabolic inhibitors or competing resistant mechanisms (e.g., Glutathione S-Transferases or GSTs are not known to favor similar intoxication routes as cytochrome P450s) [21,29,37,38]. Ultimately, as chlorfenapyr is a physiological toxin, normal mosquito behaviors during their circadian rhythms will favor intoxication [19,20] and its evaluation in small cages may yield different (poorer) results compared to experimental houses.

Novel chemistries are challenging the original 'neurotoxic thinking' of the mode of action of insecticides and are pushing testing procedures to move beyond quantification of acute mortality to account for delayed mortality and other physiological and behavioral effects. Delayed mortality has been reported for novel chemistries currently being used or evaluated for malaria IRS, clothianidin [39,40], broflanilide [17,41] and chlorfenapyr [21,34]. Delayed intoxication has also been shown for pyriproxyfen, which reduced life-span and female *Anopheles* sp. fecundity when exposed to new generation nets [42]. Our study shows for the first time the delayed mortality effect of chlorfenapyr on exposed *Ae. aegypti*. Not considering delayed mortalities may lead to considering the molecule's efficacy to be shorter than it actually is (in our case, 5 months instead of 7). This aspect was noted in an IRS WHOPES phase III trial

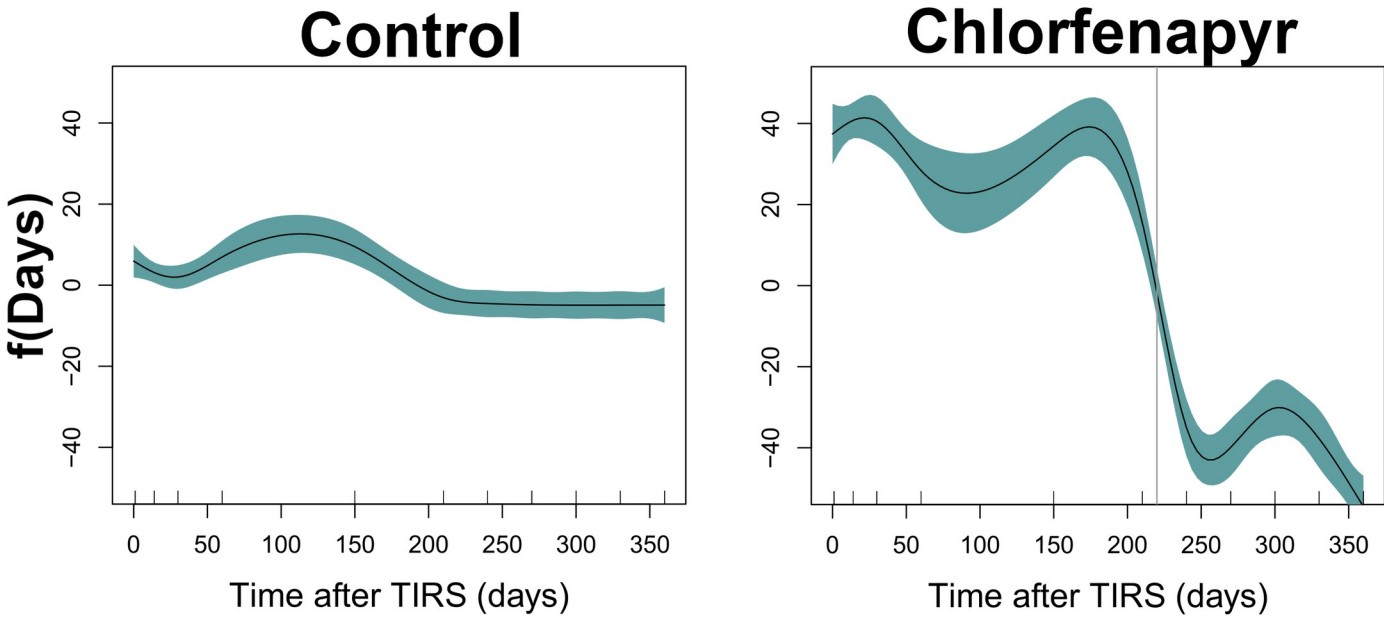

**Fig 5. Generalized Additive Linear Mixed Model (GAMM) fitted to the association between mortality [s(Mortality)] and days since TIRS application [f (Days)] for the control and chlorfenapyr houses.** The gray vertical line on the right panel shows the threshold of change from positive to negative impact of chlorfenapyr on mosquito mortality.

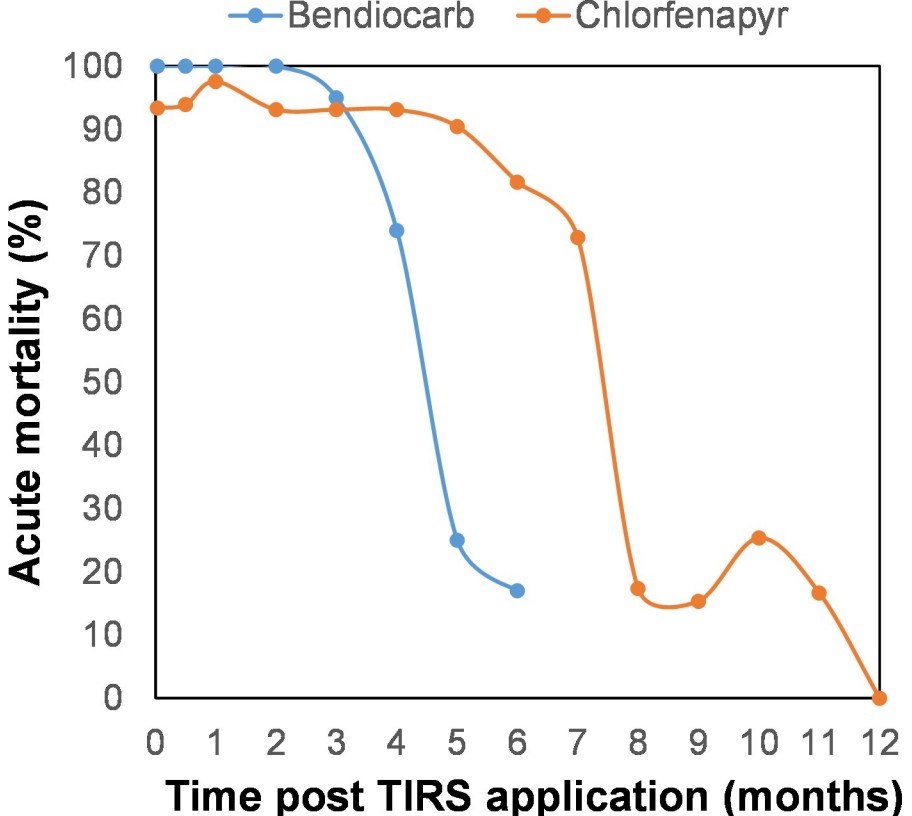

**Fig 6. Comparison of acute mortality after TIRS application of bendiocarb (Dunbar et al. [7], blue line) and chlorfenapyr (orange, present study) in experimental houses from Merida, Mexico.**

conducted in the Gambia where researchers observed that although the threshold for standard mortality metrics were observed to be declining, there was indeed a broader epidemiological impact of chlorfenapyr IRS compared to DDT [24]. Our delayed mortalities >80% of up to 7 months were comparable to WHOPES phase II hut trials against *Anopheles sp* mosquitoes, which showed 8 months efficacy after accounting for delayed mortality [24].

Some studies have demonstrated that detoxifying enzymes (P450s) in mosquitoes that are responsible for converting parent chlorfenapyr (CL303630) to its pro-insecticide metabolite (CL303268) can be inhibited with known inhibitors like PBO in measurable ways both *in vivo* and *in vitro* [29,43,44]. To less experienced researchers with this mode of action, the tendency to assume resistance rather than poor conversion (from parent to pro-insecticidal metabolite) requires consideration of laboratory or field-testing conditions which might interfere with chlorfenapyr's mode of action [29,38] as influenced by numerous endogenous and exogenous elements [21]. Other studies point to induction routes which favor pre-exposures to neuro-toxic chemistries (e.g., alpha-cypermethrin or others) which may actually enhance the conversion rates of the more toxic form of chlorfenapyr to mosquitoes as do more metabolically resistant mosquito strains (68,74). The lack of cross-resistance [43,45] and general trends for intoxication to various metabolic resistant dipterans [46] makes chlorfenapyr relevant for insecticide resistance management.

Having demonstrated utility and regional acceptance [47], the TIRS application method may provide important public health benefits when applied preventively before the transmission season [12]. Such benefit relies on the availability of long-lasting residual insecticides. Mathematical modeling showed that effectiveness of TIRS can be increased up to 90% compared to not conducting TIRS when residual efficacy of the insecticide lasts 5 months [12]. An ongoing Phase III two-arm clinical trial is evaluating the epidemiological impact of preventive TIRS on *Aedes*-borne viruses [9] using insecticides to which *Ae. aegypti* is susceptible. In urban tropical environments, pyrethroids such as deltamethrin have residual efficacies of up to 3–6 months but are severely challenged by the presence of resistance in the mosquito population [26]. Alternative chemistries (to which *Ae. aegypti* is susceptible) exist, and the carbamate bendiocarb has provided not only to control pyrethroid-resistant *Ae. aegypti* [26] but also to exert mortalities >80% for up to 4 months in experimental houses [7]. Our study shows that, in experimental houses, chlorfenapyr can extend TIRS residual efficacy against *Ae. aegypti* up to seven months (Fig 6). Future studies should evaluate the entomological impact of chlorfenapyr TIRS against *Ae. aegypti* in field randomized trials, providing evidence of the value of this new chemistry for the management of pyrethroid resistance and the prevention of *Aedes*-borne viruses.

## Supporting information

**S1 Table. Results from a Generalized Linear Mixed Model (GLMM) quantifying the significance in delayed mortality between control and treatment measures (control used as baseline).**
(DOCX)

## Acknowledgments

The authors would like to thank Yolanda Carolina Carmona Carballo, Suemy Analí Gutiérrez Martín, Eduardo José Geded Moreno, and Ana Laura Marrufo Tamayo for their dedication and efforts. Thanks are extended to Drs. James W. Austin of BASF Corporation and Susanne Stutz of BASF SE, Professional & Specialty Solutions division for providing Sylando 240SC for this trial and for consultation on unique properties of chlorfenapyr prior to study initiation.

## Author Contributions

**Conceptualization:** Azael Che-Mendoza, Fabián Correa-Morales, Oscar D. Kirstein, Pablo Manrique-Saide, Gonzalo M. Vazquez-Prokopec.

**Data curation:** Azael Che-Mendoza.

**Formal analysis:** Azael Che-Mendoza, Oscar D. Kirstein, Gonzalo M. Vazquez-Prokopec.

**Funding acquisition:** Gonzalo M. Vazquez-Prokopec.

**Investigation:** Gabriela González-Olvera, Anuar Medina-Barreiro, Carlos Arisqueta-Chablé, Wilberth Bibiano-Marin, Fabián Correa-Morales, Pablo Manrique-Saide, Gonzalo M. Vazquez-Prokopec.

**Methodology:** Azael Che-Mendoza, Gabriela González-Olvera, Carlos Arisqueta-Chablé, Wilberth Bibiano-Marin, Fabián Correa-Morales, Oscar D. Kirstein, Pablo Manrique-Saide, Gonzalo M. Vazquez-Prokopec.

**Resources:** Anuar Medina-Barreiro, Carlos Arisqueta-Chablé, Wilberth Bibiano-Marin, Fabián Correa-Morales.

**Writing – original draft:** Azael Che-Mendoza, Gabriela González-Olvera, Carlos Arisqueta-Chablé, Pablo Manrique-Saide, Gonzalo M. Vazquez-Prokopec.

**Writing – review & editing:** Pablo Manrique-Saide, Gonzalo M. Vazquez-Prokopec.

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
