## [Decision Letter · Decision Letter 0]

11 Aug 2021

Dear Dr. Vazquez-Prokopec,

Thank you very much for submitting your manuscript "Efficacy of the pyrrole insecticide chlorfenapyr against pyrethroid resistant Aedes aegypti" for consideration at PLOS Neglected Tropical Diseases. As with all papers reviewed by the journal, your manuscript was reviewed by members of the editorial board and by several independent reviewers. The reviewers appreciated the attention to an important topic. Based on the reviews, we are likely to accept this manuscript for publication, providing that you modify the manuscript according to the review recommendations. 

Sincerely,

Philip J McCall, PhD

Associate Editor

Nigel Beebe

Deputy Editor

Reviewer's Responses to Questions

**Key Review Criteria Required for Acceptance?**

**Methods**

-Are the objectives of the study clearly articulated with a clear testable hypothesis stated?

-Is the study design appropriate to address the stated objectives?

-Is the population clearly described and appropriate for the hypothesis being tested?

-Is the sample size sufficient to ensure adequate power to address the hypothesis being tested?

-Were correct statistical analysis used to support conclusions?

-Are there concerns about ethical or regulatory requirements being met?

Reviewer #1: The objective of this study is simple, and clearly stated – to evaluate the residual efficacy of Sylando TIRS against field derived Aedes aegypti. There are just a few suggestions to improve description of the methods.

One major point in the Methods which is very unclear is the replication – how many experimental houses were used in the study? In some places it seems like one pair of houses was used, one for Sylando and one for control, and in some places it seems like multiple pairs of houses were used. I think, from Figure 2, 2 houses were used, and at each time point 3 releases were done on sequential days in these 2 houses, is that right? The figure and the text should be adapted to make this clear.

When describing the results in the Abstract and in the Discussion it is important to note that the mosquitoes used for the experiment were a recently colonised lab strain, derived from local collections 4 generations previously. This is important because the fact that wild mosquitoes were not used is a minor limitation of the study, particularly because the rate of conversion of chlorfenapyr is different in different populations of mosquito and may affect the results. Releases were performed over 12 months – was a new collection made and reared to F4 each time, or was it the same colony, which would therefore have been much older than F4 by the end of the experiment? 

It is not clear in the ‘Results’ section of the Abstract whether the reference to 48h means after 48h of exposure or scored 48h after exposure. From Figure 2 I think it’s the latter, but it is not totally clear.

Line 73 – I would describe this study as ‘semi-field’ not ‘field’.

Generally the Methods are well written and clear, and I like the map and photos in Figure 1 very much.

Lines 205-206 – ‘An evaluation for these cohorts was extended up to seven days’ - this wording is very unclear, and I think you mean ‘held mosquitoes for 7 days after recapture to score delayed mortality’?

Table 1 – instead of the superscript footnotes, I suggest explaining these points in a more full table legend. Also, it is correct that exactly 100 mosquitoes were released in each experimental release? Even though that is what the method asked for, I would be surprised if exactly 100 were released each time – this has an impact when you use the number released to calculate mortality.

Equation on line 249-250 – explain all terms.

Reviewer #2: (No Response)

**Results**

-Does the analysis presented match the analysis plan?

-Are the results clearly and completely presented?

-Are the figures (Tables, Images) of sufficient quality for clarity?

Reviewer #1: The analysis for Figure 5 could be better described so that it can be interpreted more easily, but otherwise the results are analysed and presented well.

Lines 269-270 – 3 values are given for 5 time points, but surely there should be 5?

Lines 271-273 – this is not clearly described, which mosquitoes were used to test susceptibility, a subset of that same generation? Was this done at each time point, or just at the start of the experiment.

Line 305 – be more specific about the time when mortality was scored than ‘delayed mortality’ – was this 7 days post-exposure?

Figure 5 – This is not clear to me, and it would be good to explain a bit more. When you say there was a ‘reduction in mortality’, reduction compared to what? I don’t understand where the negative values come from, or what ‘lower mortality than expected under the hypothesis of no association’ means.

Reviewer #2: (No Response)

**Conclusions**

-Are the conclusions supported by the data presented?

-Are the limitations of analysis clearly described?

-Do the authors discuss how these data can be helpful to advance our understanding of the topic under study?

-Is public health relevance addressed?

Reviewer #1: The conclusion in the first paragraph of the Discussion is very clear in scope and significance, and supported by the data from this study. Some of the other statements of Conclusion are less well supported, and sentences in the Discussion are not so clear, as described below.

In the last sentence of the abstract, I would like to see more explanation of how this study shows that chlorfenapyr TIRS would be ‘feasible’ in these circumstances.

Lines 408-410 – you have not really discussed ‘large scale entomological impacts of chlorfenapyr’, what might these be, beyond delayed mortality?

Lines 111-114 –The claimed link between the mode of action of chlorfenapyr and insecticide relevance mechanisms should be explained.

Lines 399-400 – you state the efficacy can be increased by 90% when IRS lasts 5 months, but compared to what?

Reviewer #2: (No Response)

**Editorial and Data Presentation Modifications?**

Reviewer #1: Line 96 – ‘IRS’ not ‘ISS’?

All statements in lines 347-364 need to be supported by references.

Lines 369-370, for readers who do not know, please state the significance of these 4 insecticides.

Line 378 – I think you mean ‘>80%’ not <80%’?

Lines 381-394 – this paragraph is not linked to the current experiment, and I don’t see the value in including this text.

Reviewer #2: (No Response)

**Summary and General Comments**

Reviewer #1: This manuscript very nicely presents a solid experiment which demonstrates that Sylando, an IRS formulation of chlorfenapyr, is effective for 7 months when used in a TIRS strategy in semi-field experimental conditions. The method has been published previously, but the demonstration of the improved performance of this formulation against Aedes aegypti is a valuable novel finding. The study is clearly described, with just a few points of clarification needed, and a careful edit to correct some minor English errors.

I think that TIRS should be mentioned in the title, since the mode of delivering the insecticide is part of the novelty and interest of this paper.

Reviewer #2: Dear Authors,

Thank you for submitting this interesting manuscript.

STRENGTHS OF THE MANUSCRIPT

The study design is simple, results are unambiguous and clearly reported. 

Chlorfenapyr is a new product in vector control and due to its late acting properties, it needs the assessment of delayed mortality, something that is not common in this kind of studies. 

Sylando 240 SC shows a good potential as an IRS product to overcome insecticide resistance 

MAJOR COMMENTS

A positive control like a pyrethroid or a carbamate used in IRS would have added strength to the study. 

MINOR COMMENTS for Authors

Authors describe a “replicated system of two experimental houses”, Lines 132-133, with distance between them ranging “from 0.3 to 2 km” Lines 136. I read this as 4 houses with 2 treatments and 2 controls, otherwise the distance between 2 houses would be a single value. On the other hand, from Figure 2 and Table 1 it seems the experimental houses were 2 only, according to the number of mosquitoes released per each time point (300+300). I suggest to better explain this in the text.

In the manuscript it is not reported where mosquitoes were kept after being collected from the experimental houses. In light of the high delayed mortality recorded, this is an important information for understanding if mosquitoes had the possibility to fly and be active after chlorfenapyr exposure, continuing to convert chlorfenapyr to its metabolite.

“where furniture could not be removed (e.g., built-in kitchen or closet cabinets) it was sealed with window screening” Lines 157-158. I understand here that mosquitoes could not access the inner parts or land on furniture, but I wonder if they could still fly under or behind it, exploiting the gaps with walls and floors. Maybe it would be worth to explain this. 

Please, check the following sentences as it looks to me that Methods do not match with Results:

Lines 209-212 “On each house, we placed three unsprayed control cups (250 mL) containing 10 JP strain females each during the 24-h exposure period to have an independent measure of mosquito mortality due to the temperature and humidity conditions of the experimental houses. This measure was estimated at the +5, +7, +8-12 months post-application…”; and Lines 268-271 “Mortality within cups left inside houses to monitor natural mortality averaged 4.4 ± 1.3%, 1.5 ± 0.7% and 5.0 ± 1.7% (Mean ± SEM) for evaluations from +8, +9, +10, +11 and +12 months post-application, respectively…”

Check for typos Lines 160 and 280.

Line 345. It is the first time MoH appears in the manuscript. I suggest writing it in extenso.

Sincerely

PLOS authors have the option to publish the peer review history of their article (what does this mean?). If published, this will include your full peer review and any attached files.

Reviewer #1: Yes: Rosemary Susan Lees

Reviewer #2: No

Figure Files:

Data Requirements:

Reproducibility:

References

---

## [Decision Letter · Decision Letter 1]

19 Sep 2021

Dear Dr. Vazquez-Prokopec,

We are pleased to inform you that your manuscript 'Efficacy of targeted indoor residual spraying with the pyrrole insecticide chlorfenapyr against pyrethroid-resistant Aedes aegypti' has been provisionally accepted for publication in PLOS Neglected Tropical Diseases.

Best regards,

Philip J McCall, PhD

Associate Editor

Nigel Beebe

Deputy Editor

Reviewer's Responses to Questions

**Key Review Criteria Required for Acceptance?**

**Methods**

-Are the objectives of the study clearly articulated with a clear testable hypothesis stated?

-Is the study design appropriate to address the stated objectives?

-Is the population clearly described and appropriate for the hypothesis being tested?

-Is the sample size sufficient to ensure adequate power to address the hypothesis being tested?

-Were correct statistical analysis used to support conclusions?

-Are there concerns about ethical or regulatory requirements being met?

Reviewer #1: (No Response)

Reviewer #2: (No Response)

**Results**

-Does the analysis presented match the analysis plan?

-Are the results clearly and completely presented?

-Are the figures (Tables, Images) of sufficient quality for clarity?

Reviewer #1: (No Response)

Reviewer #2: (No Response)

**Conclusions**

-Are the conclusions supported by the data presented?

-Are the limitations of analysis clearly described?

-Do the authors discuss how these data can be helpful to advance our understanding of the topic under study?

-Is public health relevance addressed?

Reviewer #1: (No Response)

Reviewer #2: (No Response)

**Editorial and Data Presentation Modifications?**

Reviewer #1: (No Response)

Reviewer #2: (No Response)

**Summary and General Comments**

Reviewer #1: I am happy that the comments of both reviewers have been addressed in full. These edits have clarified the areas of uncertainty and provided useful additional detail. I believe it is now acceptable for publication.

Reviewer #2: I thank the Authors for addressing my comments. The manuscript has improved and it reads well. I have no further suggestions.

Regards

PLOS authors have the option to publish the peer review history of their article (what does this mean?). If published, this will include your full peer review and any attached files.

Reviewer #1: **Yes: **Rosemary Susan Lees

Reviewer #2: **Yes: **Luca Facchinelli

---

## [Editor Report · Acceptance letter]

30 Sep 2021

Dear Dr. Vazquez-Prokopec,

We are delighted to inform you that your manuscript, " Efficacy of targeted indoor residual spraying with the pyrrole insecticide chlorfenapyr against pyrethroid-resistant Aedes aegypti ," has been formally accepted for publication in PLOS Neglected Tropical Diseases.

Best regards,

Shaden Kamhawi

co-Editor-in-Chief

Paul Brindley

co-Editor-in-Chief
